# Beyond Simple Sum of Delayed Rewards: Non-Markovian Reward Modeling for Reinforcement Learning

## Abstract

Reinforcement Learning (RL) empowers agents to acquire various skills by learning from reward signals. Unfortunately, designing high-quality instance-level rewards often demands significant effort. An emerging alternative, RL with delayed reward, focuses on learning from rewards presented periodically, which can be obtained from human evaluators assessing the agent's performance over sequences of behaviors. However, traditional methods in this domain assume the existence of underlying Markovian rewards and that the observed delayed reward is simply the sum of instance-level rewards, both of which often do not align well with real-world scenarios. In this paper, we introduce the problem of RL from Composite Delayed Reward (RLCoDe), which generalizes traditional RL from delayed rewards by eliminating the strong assumption. We suggest that the delayed reward may arise from a more complex structure reflecting the overall contribution of the sequence. To address this problem, we present a framework for modeling composite delayed rewards, using a weighted sum of non-Markovian components to capture the different contributions of individual steps. Building on this framework, we propose Composite Delayed Reward Transformer (CoDeTr), which incorporates a specialized in-sequence attention mechanism to effectively model these contributions. We conduct experiments on challenging locomotion tasks where the agent receives delayed rewards computed from composite functions of observable step rewards. The experimental results indicate that CoDeTr consistently outperforms baseline methods across evaluated metrics. Additionally, we demonstrate that it effectively identifies the most significant time steps within the sequence and accurately predicts rewards that closely reflect the environment feedback. Code is available at an anonymous link: https://anonymous.4open.science/r/CoDe-67E8/.

## 1 Introduction

Reinforcement Learning (RL) has become a powerful paradigm for solving sequential decision-making problems by enabling agents to learn optimal policies through interactions with their environments (Kaelbling et al., 1996; Arulkumaran et al., 2017). A critical component of RL is the reward function, which guides the learning process by indicating the desirability of different states and actions. In complex real-world applications such as autonomous driving (Sallab et al., 2017; Kiran et al., 2021), financial trading (Yang et al., 2020; Hambly et al., 2023), and healthcare (Coronato et al., 2020; Yu et al., 2021), designing detailed reward signals for every possible state-action pair is often impractical due to the high dimensionality and complexity of these domains.

To address the challenge of specifying fine-grained rewards, researchers have explored the use of delayed rewards. Instead of assigning rewards at every individual step or action, feedback is given based on the outcome of a sequence of actions (Liu et al., 2019; Gangwani et al., 2020; Efroni et al., 2021; Raposo et al., 2021; Ren et al., 2021). These works generally aim to simplify the reward design process while still allowing for effective policy learning. For example, Liu et al. (2019) leveraged sequence modeling with expert demonstrations, while Gangwani et al. (2020) and Ren et al. (2021) proposed iterative and randomized approaches to refine reward credit assignment. While these approaches reduce the burden of reward engineering, they typically rely on two key assumptions. First, they assume the existence of underlying *Markovian* rewards, meaning that the

Figure 1: Illustration of our framework. The Composite Delayed Reward Transformer generates the predicted non-Markovian rewards $\hat{r}_t$ along with the corresponding importance weights $w_t$ for each sequence. The final composite reward for each sequence is calculated as a weighted sum of the predicted rewards.

reward at any step depends solely on the current state and action. Second, they posit that the structure of the delayed reward is a *simple sum* of individual rewards associated with each state-action pair, with equal weighting across the sequence.

However, these assumptions may not hold in many practical scenarios. The Markovian assumption neglects the fact that rewards can depend on sequences of states or actions that are not fully captured by the current state (Bacchus et al., 1996; 1997). Furthermore, the assumption of equal-weighted summation does not align with how humans evaluate experiences. Psychological studies have shown that people tend to assign disproportionate importance to remarkable or significant moments within an experience (Kahneman, 2000; Newell et al., 2022). This suggests that in tasks involving human feedback, certain states or actions within a sequence may contribute more heavily to the overall assessment than others. An example of this phenomenon is observed in high-stake environments studied by Klein (2008): experts such as firefighters, pilots, and emergency medical personnel often focus intensely on critical cues and pivotal moments that can significantly affect outcomes. These professionals rely on recognizing patterns and key indicators to make rapid decisions, effectively assigning greater weight to crucial information rather than treating all information equally. Therefore, previous methods that assume equal-weighted summation of Markovian rewards may fail to capture the true nature of the feedback. As a result, these methods may not yield promising results in scenarios where the reward structure is inherently non-Markovian and where critical moments disproportionately influence the overall evaluation.

Building on the above observations, we emphasize the need for RL frameworks that can address the RL from Composite Delayed Reward (RLCoDe) problem by accommodating non-Markovian rewards and capturing the disproportionate weighting inherent in delayed feedback. To meet this need, we propose the Composite Delayed Reward Transformer (CoDeTr), which introduces a non-Markovian reward model that predicts rewards more accurately reflecting the underlying reward structure as perceived by humans. Additionally, CoDeTr incorporates an *in-sequence attention mechanism* to model the reward aggregation process, thereby capturing critical instances within a sequence and reflecting the varying importance that human feedback assigns to these moments. By integrating these non-Markovian rewards into the policy learning process, the agent can learn more effectively in environments where traditional assumptions about rewards do not hold.

Our main contributions can be summarized as follows:

- We identify the limitations of existing delayed reward frameworks relying on Markovian assumption and equal-weighted summation, and proposed RLCoDe to address these issues.

- We propose CoDeTr to accommodate non-Markovian reward functions by capturing the disproportionate weight of feedback through an in-sequence attention mechanism.

- We demonstrate that our approach outperforms state-of-the-art delayed reward methods in environments where rewards depend on the sequence of visited states and where critical moments have a greater impact on the overall evaluation.

- We verify that our method effectively learns the rewards corresponding to the agent's actions, while also identifying which specific steps within the sequence are given more emphasis by the composite delayed reward.

## 2 PRELIMINARIES

In this section, we revisit conventional RL and RL from delayed rewards under the assumption of Markovian rewards. Building on these concepts, we introduce the problem of RL from Composite Delayed Reward (RLCoDe), which generalizes these settings by removing the Markovian assumption and allowing for non-Markovian reward structures with flexible, non-uniform weighting of contributions across the sequence.

### 2.1 STANDARD REINFORCEMENT LEARNING

In standard RL (Bellman, 1966), the environment is modeled as a Markov Decision Process (MDP):

**Definition 1 (MDP)** *An MDP is defined by the tuple $\mathcal{M} = (\mathcal{S}, \mathcal{A}, P, r, \mu)$, where*

- *$\mathcal{S}$ is a finite set of states.*

- *$\mathcal{A}$ is a finite set of actions.*

- *$P : \mathcal{S} \times \mathcal{A} \times \mathcal{S} \to [0, 1]$ is the state transition probability function, where $P(\mathbf{s}'|\mathbf{s}, \mathbf{a})$ gives the probability of transitioning to state $\mathbf{s}'$ from state $\mathbf{s}$ after taking action $\mathbf{a}$.*

- *$r : \mathcal{S} \times \mathcal{A} \to \mathbb{R}$ is the immediate reward function.*

- *$\mu$ is the initial state distribution.*

An agent interacts with the environment by following a policy $\pi : \mathcal{S} \times \mathcal{A} \to [0, 1]$, where $\pi(\mathbf{a}|\mathbf{s})$ is the probability of taking action $\mathbf{a}$ in state $\mathbf{s}$. The objective is to find an optimal policy $\pi^*$ that maximizes the expected cumulative reward:

$$J(\pi) = \mathbb{E}_\pi \left[ \sum_{t=0}^{\infty} r(\mathbf{s}_t, \mathbf{a}_t) \right], \tag{1}$$

where the expectation is over trajectories induced by the policy $\pi$ and the transition dynamics $P$.

### 2.2 REINFORCEMENT LEARNING FROM DELAYED REWARDS

In many real-world applications, immediate rewards are not readily available or are difficult to specify (Devidze et al., 2022; Tang et al., 2024). Instead, the agent may receive delayed rewards, often at the end of a sequence or trajectory. This setting is common in domains where the outcome of actions is not immediately observable or when feedback is provided by human evaluators who assess performance over extended periods (Shen & Chi, 2016; Krishnan et al., 2019; Gao et al., 2024).

In RL with delayed rewards, the environment is modeled as an MDP with a cumulative reward $R(\tau)$ provided for a sequence $\tau$ or trajectory $\mathcal{T}$. Let $\mathcal{T} = \{(\mathbf{s}_0, \mathbf{a}_0), (\mathbf{s}_1, \mathbf{a}_1), \ldots, (\mathbf{s}_{T-1}, \mathbf{a}_{T-1})\}$ represent an agent trajectory over $T$ time steps, encompassing all experienced states and actions within an episode. A sequence $\tau$ refers to a portion of the trajectory $\mathcal{T}$, starting at time step $i$ and consisting of $n_i$ state-action pairs:

$$\tau = \{(\mathbf{s}_i, \mathbf{a}_i), (\mathbf{s}_{i+1}, \mathbf{a}_{i+1}), \ldots, (\mathbf{s}_{i+n_i-1}, \mathbf{a}_{i+n_i-1})\}.$$

A common assumption is that there exists an underlying Markovian reward function $r(\mathbf{s}, \mathbf{a})$ such that the sequence-level reward can be decomposed as

$$R(\tau) = \sum_{t=i}^{i+n_i-1} r(\mathbf{s}_t, \mathbf{a}_t). \tag{2}$$

This assumption simplifies the problem by allowing standard RL algorithms to be applied after redistributing the cumulative reward $R(\tau)$ back to individual time steps. However, this approach

may not be suitable in situations where the reward at each time step depends on the context of the entire sequence, or when the delayed reward places greater emphasis on certain critical time steps.

## 2.3 Reinforcement Learning from Composite Delayed Reward

To address the limitations of assuming Markovian and additive reward structure, we introduce the Composite Delayed Reward Markov Decision Process (CoDeMDP), which generalizes the conventional MDP to allow for non-Markovian rewards and non-uniform weighting of contributions.

**Definition 2 (CoDeMDP)** *A CoDeMDP is defined by the tuple $(\mathcal{S}, \mathcal{A}, P, R_{co}, \mu)$, where*

- $\mathcal{S}$ *is the set of states.*

- $\mathcal{A}$ *is the set of actions.*

- $P : \mathcal{S} \times \mathcal{A} \times \mathcal{S} \to [0, 1]$ *is the state transition probability function.*

- $R_{co} : \tau \to \mathbb{R}$ *is the composite delayed reward function, defined on sequences $\tau$ within a trajectory $\mathcal{T}$.*

- $\mu$ *is the initial state distribution.*

This composite delayed reward is assigned periodically, based on the sequence of states and actions between two reward points, similar to traditional delayed reward settings (Gangwani et al., 2020; Ren et al., 2021). In this framework, $R_{co}$ is a reward function of the entire sequence $\tau$, which may depend on complex interactions among states and actions, without assuming Markovian properties for instance-level rewards or requiring the sequence-level reward to be additive by instance-level rewards. This generalization allows us to model scenarios where the reward depends on the sequence history, future outcomes, or non-linear aggregation of individual contributions. The agent's objective is to maximize the expected cumulative composite reward over trajectories:

$$J(\pi) = \mathbb{E}_{\mathcal{T}(\pi)} \left[ \sum_{\tau \subseteq \mathcal{T}} R_{co}(\tau) \right], \tag{3}$$

where $\mathcal{T}(\pi)$ denotes the distribution over trajectories induced by policy $\pi$.

## 3 Proposed Method

In this section, we introduce our proposed method to address the challenges of RLCoDe. We begin by presenting a general framework for modeling delayed rewards using a weighted sum of non-Markovian components. Following this, we propose a transformer-based architecture called CoDeTr, specifically designed to predict instance-level non-Markovian rewards and integrate them into a sequence-level composite delayed reward.

### 3.1 Sequence-level Reward Decomposition

As discussed above, most prior work assumes that the instance-level reward is Markovian and that sequence-level feedback is based on an equal-weighted sum of immediate rewards. This assumption may not hold in many real-world scenarios if the reward depends on the sequence of states and actions, including historical context and future outcomes, rather than solely on the current state and action (Bacchus et al., 1996; 1997; Early et al., 2022). Meanwhile, some certain states or actions within a sequence have a disproportionate impact on the overall reward, reflecting the human tendency to assign greater importance to critical moments (Kahneman, 2000; Newell et al., 2022). These limitations necessitate a framework that can capture complex reward dependencies and varying importance of different parts of a trajectory. To address these challenges, we propose modeling the sequence-level reward using a non-Markovian reward function $\hat{r}$ with learnable weights $w$:

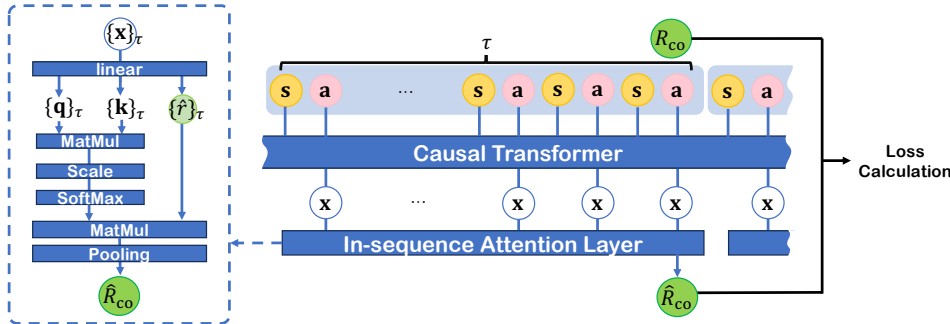

Figure 2: The architecture of the proposed Composite Delayed Reward Transformer. The model processes the sequence of state-action pairs using a causal transformer, where the embeddings $x$ represent the context information from the initial time step to current time step. The in-sequence attention mechanism computes the non-Markovian rewards $\{\hat{r}\}_\tau$, the queries $\{\mathbf{q}\}_\tau$, and keys $\{\mathbf{k}\}_\tau$, in a sequence $\tau$. The query and key vectors are multiplied and passed through a softmax operation to compute attention weights. The attention-weighted sum of instance-level rewards is then aggregated via sum pooling to generate the final sequence-level reward $\hat{R}_{\mathrm{co}}(\tau)$ to approximate $R_{co}(\tau)$.

$$\hat{R}_{\mathrm{co}}(\tau) = \sum_{t=i}^{i+n_i-1} w\big(\{(\mathbf{s}_{t'}, \mathbf{a}_{t'})\}_{t'=i}^{i+n_i-1}; \psi\big)_t \cdot \hat{r}\big(\{(\mathbf{s}_{t'}, \mathbf{a}_{t'})\}_{t'=i}^{t}; \psi\big)_t, \tag{4}$$

where $\psi$ represents the learnable parameters for both the weight function $w$ and the reward function $\hat{r}$, ensuring they can be jointly optimized for more effective learning of the reward structure. Unlike traditional Markovian rewards that depend only on the current state and action, $\hat{r}$ considers the historical sequence of state-action pairs $\{(\mathbf{s}_{t'}, \mathbf{a}_{t'})\}_{t'=i}^{t}$ up to the current time step $t$. The weight function $w$ takes the entire sequence $\tau = \{(\mathbf{s}_{t'}, \mathbf{a}_{t'})\}_{t'=i}^{i+n_i-1}$ as input, allowing the model to evaluate the importance of each time step in the context of the whole sequence. This formulation defines how the sequence-level reward is related to instance-level contributions and their associated weights.

## 3.2 ARCHITECTURE

To implement the proposed approach, we design a transformer-based architecture called Composite Delayed Reward Transformer (CoDeTr), which functions as a reward model for predicting instance-level non-Markovian rewards and representing the composite delayed reward.

**Instance-Level Reward Prediction.** We adopt the transformer network (Vaswani et al., 2017) as the backbone for instance-level reward prediction. Specifically, we employ the GPT architecture (Radford et al., 2018), which utilizes a causal self-attention mechanism. This causal transformer ensures the chronological order of state-action pairs is preserved in our proposed reward model. For each time step $t$ in a sequence of $M$ time steps, the causal transformer, represented as a function $g$, processes the input sequence $\sigma = \{(\mathbf{s}_0, \mathbf{a}_0), \ldots, (\mathbf{s}_{M-1}, \mathbf{a}_{M-1})\}$, generating outputs $\{\mathbf{x}_t\}_{t=0}^{M-1} = g(\sigma)$. By aligning the output $\mathbf{x}_t$ with the action token $\mathbf{a}_t$, we directly model the consequences of actions, which are pivotal in computing immediate rewards and predicting subsequent states, thereby helping the model better understand environmental dynamics.

After obtaining the output embeddings $\{\mathbf{x}_t\}_{t=0}^{M-1}$ from the causal transformer, a linear transformation is applied to each $\mathbf{x}_t$ to compute the corresponding instance-level reward $\hat{r}_t$:

$$\hat{r}_t = \mathrm{Linear}(\mathbf{x}_t). \tag{5}$$

Since the underlying instance-level reward depends on the previous state-action pairs, this allows us to model the immediate rewards associated with each action in the sequence, capturing the essential dynamics at the instance level.

**Composite Delayed Reward Representation.** To represent the composite delayed reward based on instance-level rewards, we use an in-sequence attention mechanism inspired by traditional attention models (Vaswani et al., 2017). The in-sequence attention mechanism operates only within the sequence $\tau$ corresponding to the composite delayed reward, ensuring that the attention mechanism focuses solely on the instance-level rewards that are relevant to the specific sequence. This attention is bi-directional, allowing the model to consider the entire sequence when determining the importance of each time step.

Specifically, we apply two linear transformations to each embedding $\mathbf{x}_t$, resulting in query embeddings $\mathbf{q}_t \in \mathbb{R}^d$ and key embeddings $\mathbf{k}_t \in \mathbb{R}^d$, where $d$ is the embedding dimension. The attention weight for each time step is calculated using the dot product between the query and key embeddings, followed by a softmax operation to normalize the weights:

$$\hat{R}_{\text{co}}(\tau) = \sum_{i \in \tau} \sum_{t \in \tau} \text{softmax} \left( \frac{\{\langle \mathbf{q}_i, \mathbf{k}_{t'} \rangle\}_{t' \in \tau}}{\sqrt{d}} \right) \hat{r}_t, \tag{6}$$

where $\langle \cdot, \cdot \rangle$ denotes the dot product, and the scaling factor $\sqrt{d}$ is used to prevent extremely small gradients (Vaswani et al., 2017). The importance weight for each time step $t$ is given by:

$$w_t = \sum_{i \in \tau} \text{softmax} \left( \frac{\{\langle \mathbf{q}_i, \mathbf{k}_{t'} \rangle\}_{t' \in \tau}}{\sqrt{d}} \right). \tag{7}$$

In this formulation, the attention mechanism captures the relationships among all instances within the sequence, aligning with the requirements of sequence-level reward prediction. By assigning different weights to each time step, the model can focus on critical moments that disproportionately influence the overall evaluation.

### 3.3 Learning Process

We train the proposed reward model by minimizing the loss between the observable composite reward $R_{\text{co}}(\tau)$ and the predicted reward $\hat{R}_{\text{co}}(\tau)$ using the mean square error $(R_{\text{co}}(\tau) - \hat{R}_{\text{co}}(\tau))^2$. For RL, the learned reward function is used to label all state-action pairs. Since we are training a non-Markovian reward function, we provide the model with the past $H$ transitions, $\{(\mathbf{s}_{t-H+1}, \mathbf{a}_{t-H+1}), \ldots, (\mathbf{s}_t, \mathbf{a}_t)\}$, and use the output at time step $t$ from the attention layer as the reward for that time step. This approach allows the agent to learn policies that consider the sequence context and assign appropriate credit. The training process alternates between updating the reward model and training the policy: the agent generates delayed reward sequences from environment interactions to update the reward model, which then labels instance-level rewards to refine the policy. This iterative process leads to mutual improvement. See Appendix B for algorithm details.

## 4 Experiment

In this section, we begin by evaluating the empirical performance of our proposed method across a variety of benchmark tasks from MuJoCo (Todorov et al., 2012) and the DeepMind Control Suite (Tassa et al., 2018), each featuring different types and lengths of composite delayed rewards. We then assess the effectiveness of proposed method on traditional sum-form delayed rewards, examining whether removing the sum-form assumption affects training compared to established baselines. Lastly, we analyze the relationship between the predicted rewards, the learned attention weights, and the actual environment conditions, providing insights into the interpretability and accuracy of our reward model.

### 4.1 Compare with Baseline Methods

**Experiment Setting.** We evaluated our method on benchmark tasks from the MuJoCo locomotion suite (Ant-v2, HalfCheetah-v2, and Walker2d-v2) and the DeepMind Control Suite (fish-upright

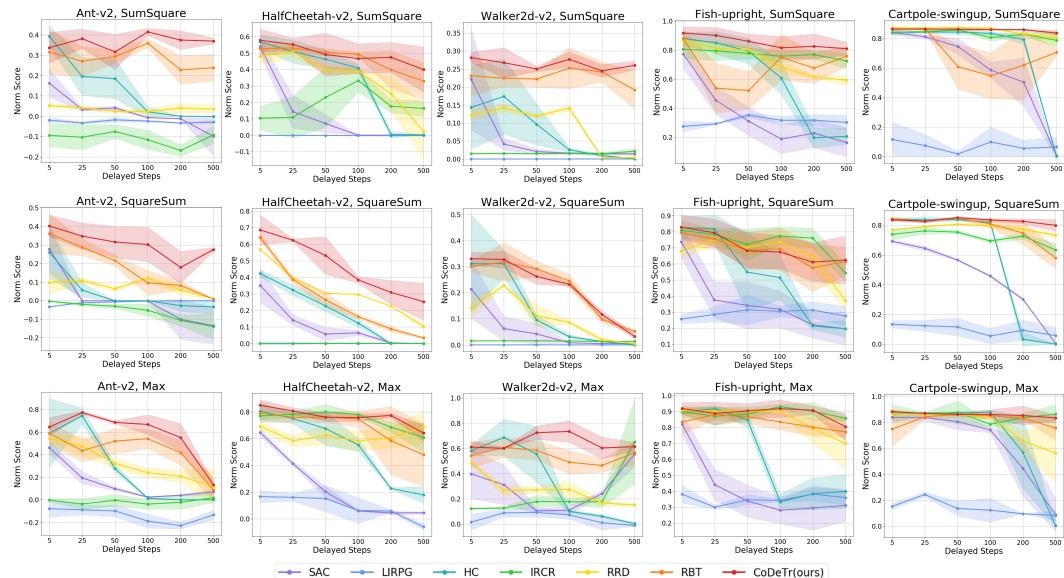

Figure 3: A performance comparison of composite delayed rewards, SumSquare (upper), Square-Sum (meddle), and Max (lower), across MuJoCo and DeepMind Control Suite environments with six different delay lengths (5, 25, 50, 100, 200, and 500). The normalized scores are averaged over 3 trials, with the mean and standard deviation computed across a total of 1e6 time steps.

and cartpole-swingup). Differing from standard environments where rewards are assigned at each step, our approach involved assigning a composite reward at the end of sequence while assigning a reward of zero to all other state-action pairs. Unlike previous work, which typically assumes that the sequence-level reward is simply the sum of individual step rewards, our composite reward encompasses more complex forms. Specifically, we included the following composite delayed reward structures, computed over a segment $\tau = \{(\mathbf{s}_t, \mathbf{a}_t)\}_{t=i}^{i+n_i-1}$, where $n_i$ represents the length of the delayed steps:

- **SumSquare**: The composite delayed reward is the sum of squared step rewards, placing more emphasis on larger rewards: $R_{\mathrm{co}} = \sum_{t=i}^{i+n_i-1} \mathrm{abs}(r_t) \cdot r_t$.

- **SquareSum**: The composite delayed reward is the square of the sum of step rewards, highlighting overall sequence performance: $R_{\mathrm{co}} = \mathrm{abs}\big( \sum_{t=i}^{i+n_i-1} r_t \big) \cdot \big( \sum_{t=i}^{i+n_i-1} r_t \big)$.

- **Max**: The composite delayed reward is a softmax-weighted sum of step rewards, giving more attention to the highest rewards: $R_{\mathrm{co}} = \sum_{t=i}^{i+n_i-1} \frac{n_i \cdot e^{\beta r_t}}{\sum_{t'=i}^{i+n_i-1} e^{\beta r_{t'}}} \cdot r_t$,

  where $\beta$ is a scaling parameter that controls the sharpness of the softmax distribution.

Here, $r_t = r(\mathbf{s}_t, \mathbf{a}_t)$ represents the unobservable Markovian reward at time step $t$ from the original environment. Additionally, we investigated the impact of varying levels of reward delay by setting the delay steps $n_i$ to 5, 25, 50, 100, 200, and 500. These composite delayed rewards complicate credit assignment to state-action pairs and capturing sequence dependencies. Longer delays weaken the correlation between actions and rewards, making it harder to trace rewards back to specific actions. Both the complexity of composite rewards and extended delays present significant challenges for effective learning. Our experiments are based on the Soft Actor-Critic (SAC) (Haarnoja et al., 2018) algorithm, with the maximum length for each episode fixed at 1000 steps across all tasks. Further experimental details are provided in the Appendix A.

**Baselines.** In the comparative analysis, our framework was rigorously evaluated against several leading algorithms in the domain of RL with delayed reward:

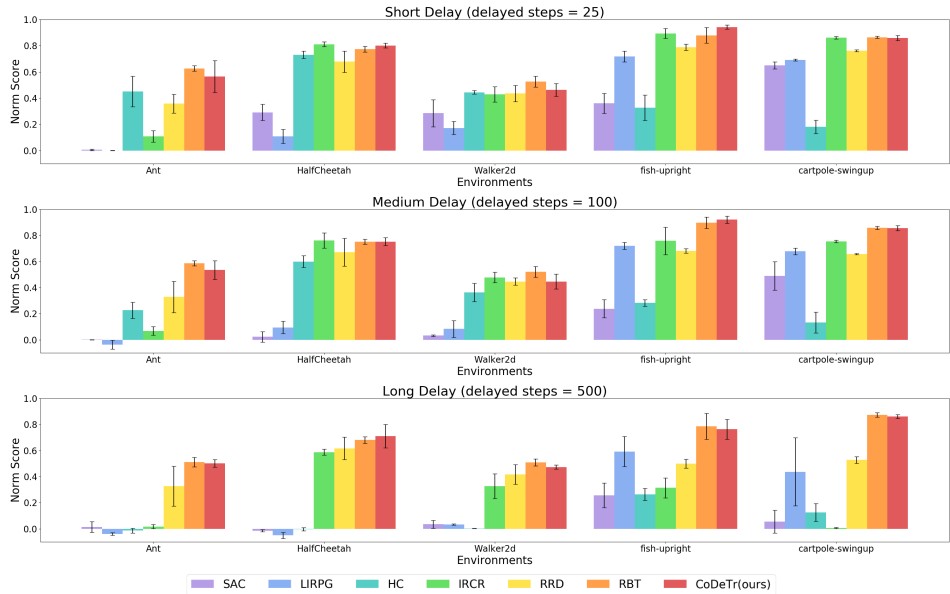

Figure 4: Performance comparison of sum-form delayed rewards in MuJoCo and DeepMind Control Suite environments with three different delay lengths. The mean and standard deviation of the normalized scores are reported over 6 trials, spanning a total of 1e6 time steps.

- **SAC** (Haarnoja et al., 2018): It directly utilized the original delayed reward information for policy training using the SAC algorithm.

- **LIRPG** (Zheng et al., 2018): It learned an intrinsic reward function to complement sparse environmental feedback, training policies to maximize combined extrinsic and intrinsic rewards. We used the same code provided by the paper.

- **HC** (Han et al., 2022): The HC-decomposition framework was utilized to train the policy using a value function that operates on sequences of data. We used the original implementation as provided by the paper.

- **IRCR** (Gangwani et al., 2020): It adopted a non-parametric uniform delayed reward redistribution. We used the code supplied by the original paper.

- **RRD** (Ren et al., 2021): It employed a reward model trained with a randomized return decomposition loss. We used the same code provided by the paper.

- **RBT** (Tang et al., 2024): It was employed under the sum-form reward with uniform weight assumption for reward redistribution, utilizing a transformer-based reward model. We employed the code as the original paper.

**Evaluation Metric.**    For evaluation, we report the normalized average accumulative reward across 3 seeds with random initialization to demonstrate the performance of evaluated methods. Higher accumulative reward in evaluation indicates better performance. Details of the normalization procedure are provided in the Appendix A.3.

**Overall Performance Comparison.**    In this part, the delayed rewards no longer align with the sum-form assumption used in previous work. In Fig. 3, the upper, medium, and lower rows shows the SumSquare, SquareSum, and Max composite delayed reward result, respectively. For SumSquare, our method consistently outperforms baseline methods across different environments, maintaining strong performance even as the delay in reward steps increases. In the SquareSum setting, the delayed reward is more complex, as it emphasizes the cumulative effect of multiple steps rather than focusing on individual large rewards, making it more difficult to attribute rewards to specific actions. Despite this complexity, our method still holds a clear advantage over most baselines, even as performance slightly decreases with longer delays. In the Max setting, where rewards are sparse and only

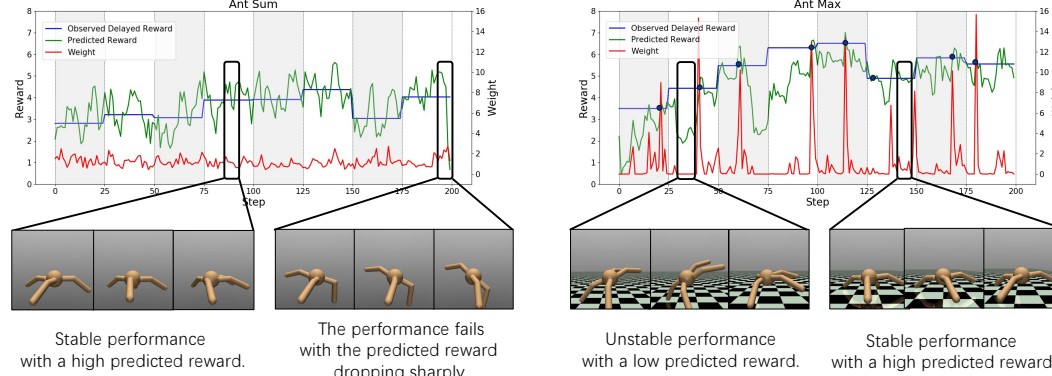

Figure 5: Comparison of mean of observed delayed rewards (blue line), predicted rewards (green line), and learned weights (red line) in the Ant environment under two different delayed reward structures: Sum (left) and Max (right). In the Max setting, the blue points indicate the steps with the highest rewards in the original environment. Every 25 steps form a delayed reward sequence, after which a composite delayed reward is assigned. The images below correspond to the behavior of agent at the time steps highlighted by the black frames in the plots.

a few key steps contribute to the overall outcome, our method manages to learn effectively, showing that it can still identify and focus on the most critical instances in the sequence, even with limited reward signals. Under these conditions, methods relying on the sum-form Markovian rewards with uniform weights assumption experience a significant drop in performance. This indicates that these methods are heavily dependent on this assumption. It is also worth noting that the HC method does not rely on this assumption, but its effectiveness is limited to shorter delays. As the delay steps increase, its performance drops sharply.

Overall, our proposed method shows a strong ability to handle various composite delayed reward structures and adapt to increasing delays, outperforming baseline methods in most environments. Moreover, these results highlight the importance of further exploring how to learn reward models and train policies in environments with composite delayed rewards, making it worthy of deeper investigation. Additional experimental results for varying delayed lengths are provided in Appendix C.

## 4.2 Performance on Traditional Delayed Reward

Fig. 4 illustrates the results of the traditional sum-form delayed reward setting, where the observed composite delayed rewards are simply the sum of the Markovian rewards provided by the original environment. It is important to note that this experiment is consistent with those conducted in prior works and aligns with the assumptions of baseline methods (Ren et al., 2021; Arjona-Medina et al., 2019; Tang et al., 2024). The figure presents results across short (25 steps), medium (100 steps), and long (500 steps) delays, showing that our method remains robust across different delay levels, performing comparably to the state-of-the-art baselines. The error bars indicate the variance across multiple runs, highlighting the stability of our approach. This demonstrates that our reward model is capable of learning useful information effectively, even without the restrictive sum-form assumption.

## 4.3 Case Study

In this section, we analyze the relationship between the predicted rewards and weights from our reward model and the real rewards from environment under Sum and Max settings of delayed reward composition. We chose these two settings because they provide contrasting scenarios for evaluating the effectiveness of our reward model.

In Fig. 5, we analyze the attention weights learned under both the sum-form and max-form delayed reward settings. For the sum-form delayed reward, where the true reward represents the sum of instance-level contributions, the learned attention weights consistently hover around 1, suggesting that our reward model effectively assigns uniform importance to each time step within the sequence.

On the other hand, in the max-form delayed reward setting, where the reward is determined by the highest instance-level contribution, the learned weights emphasize the critical points in the sequence corresponding to the highest rewards. Notably, within each sequence, the peaks of the learned weights align closely with the peaks of rewards in the true environment (blue points), demonstrating that our reward model effectively identifies the most critical instances.

As shown in the images below, when the agent performs stably in both settings, the predicted rewards are high at the corresponding time steps. Conversely, when the performance of agent deteriorates or fails, the predicted rewards drop significantly, even approaching zero. This indicates that the predicted rewards align well with the agent's behavior, confirming that our reward model has effectively learned a reasonable reward allocation.

The match between predicted rewards and agent behavior shows that our reward model closely aligns with the agent's performance. The learned attention weights capture dependencies between individual and overall rewards, demonstrating robustness in delayed and long-term dependencies. This flexibility allows the model to adapt across tasks and reward structures, making it effective in real-world scenarios where traditional Markovian assumptions fail.

## 5 RELATED WORK

**RL from Delayed Rewards.** Learning from aggregated or delayed rewards has been extensively studied, especially in situations where immediate feedback is impractical. Traditional methods rely on trajectory-level feedback, assuming that the reward is the sum of individual step rewards. For example, IRCR (Gangwani et al., 2020) and RRD (Ren et al., 2021) redistribute rewards under Markovian assumptions with equal weighting, while RUDDER (Arjona-Medina et al., 2019) uses recurrent neural networks for credit assignment. Extensions, including expert demonstrations and language models (Liu et al., 2019; Widrich et al., 2021; Patil et al., 2022), further improve precision. The Reward Bag Transformer (RBT) (Tang et al., 2024) models the sequential feedback as a bagged reward and employs a transformer to redistribute the bagged reward into instance-level rewards. Although these methods effectively redistribute rewards, they are limited by the assumption that all states contribute equally, which may not reflect the varying importance of different parts of a trajectory in many real-world tasks. Additionally, Han et al. (2022) proposed to directly modifiy RL algorithms by introducing a novel definition of the Q-function to leverage sequence-level information. However, this approach is only suitable for shorter sequence learning and struggles to adapt to more complex, long-term delayed reward scenarios.

**Transformers for RL.** Transformers (Vaswani et al., 2017) have shown significant effectiveness in RL, especially for sample-efficient and generalizable learning, as demonstrated in StarCraft (Vinyals et al., 2019; Zambaldi et al., 2019) and DMLab30 (Parisotto et al., 2020). In offline RL, transformers have been utilized for sequential modeling of RL problems (Chen et al., 2021; Janner et al., 2021; Kim et al., 2023). Luo et al. (2021) combined deep convolution transfer-learning models and inverse RL for reward function acquisition, while Zhang et al. (2023) transformed non-Markovian reward processes into Markovian ones, enhancing online interaction efficiency. In our work, we employ Transformers as reward models to capture non-Markovian reward dependencies in the composite delayed reward problem.

## 6 CONCLUSION

In this paper, we addressed the challenge of composite delayed reward structures in RL, a problem that extends beyond the traditional sum-form assumption commonly used in existing methods. We proposed an effective solution by introducing a reward model capable of flexibly handling various composite delayed reward structures, incorporating non-Markovian dependencies through an attention mechanism. Our approach consistently outperformed baseline methods across a range of environments and delay settings. While our results show significant improvements, this work opens several avenues for future research. Further exploration is warranted to better model and learn from complex reward structures in more diverse and real-world scenarios. Investigating how to scale our method to even longer delays or more intricate dependency patterns could provide deeper insights.

## 7 ETHICS STATEMENT

This research focuses on developing RL method with composite delayed rewards, with experiments conducted solely on simulated environments such as MuJoCo and DeepMind Control Suite, and without involving any direct human subjects. We acknowledge that reinforcement learning techniques can be applied to sensitive real-world scenarios like healthcare, autonomous driving, and social decision-making, which may have significant ethical implications. Notably, the use of composite delayed rewards in our work inherently reduces the granularity of reward information, thereby helping to protect individual privacy when applied to real-world scenarios. Throughout this research, we emphasize fairness and transparency in our methodologies to mitigate the risks of unintended consequences from applying the learned policies in such settings. Our model is trained and evaluated on well-established benchmarks, ensuring the reproducibility and reliability of our findings. We do not foresee any discrimination, bias, or privacy concerns arising from this research. Additionally, we adhere to all guidelines regarding dataset use, licensing, and attribution, and have disclosed any potential conflicts of interest. We believe our work aligns with the code of ethics and does not raise ethical concerns related to harmful outcomes or unethical applications.

## 8 REPRODUCIBILITY STATEMENT

We have made substantial efforts to ensure the reproducibility of our work. Specifically, the implementation details of our proposed method, including hyperparameters, training setups, and environment configurations, are provided in the Section 4 and further documented in Appendix A. The datasets and data processing steps are also thoroughly explained in the Appendix A to support replication. All algorithms, including the policy optimization with CoDeTr, are described in detail in Section 3 and further supported by pseudocode in Appendix B. Additionally, we provide an anonymous link to the downloadable source code in the abstract to facilitate the replication of our experiments. We believe that these resources collectively contribute to the reproducibility of our results.

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

## A    IMPLEMENTATION DETAILS

### A.1    BENCHMARKS WITH COMPOSITE DELAYED REWARD

In this work, we introduced a new problem setting, called composite delayed rewards, within the suite of locomotion benchmark tasks in both the MuJoCo environment and the DeepMind Control Suite. Our experiments were conducted using the OpenAI Gym platform (Brockman et al., 2016) and the DeepMind Control Suite (Tassa et al., 2018), focusing on tasks with extended horizons and a fixed maximum trajectory length of $T = 1000$. We utilized MuJoCo version 2.0 for our simulations, which can be accessed at `http://www.mujoco.org/`. MuJoCo operates under a commercial license, and we ensured full compliance with its licensing terms. Additionally, the DeepMind Control Suite, distributed under the Apache License 2.0, was used in accordance with its licensing requirements.

Experiments involving composite delayed rewards with varying delay steps (5, 25, 50, 100, 200, and 500) and different composite types (SumSquare, SquareSum, Max, and the traditional Sum) were conducted to validate the effectiveness of the proposed method. In the Max experiment, the scaling parameter $\beta$ is set to 3. To evaluate its performance, commonly used delayed reward algorithms were adapted to fit within the composite delayed reward framework, acting as baselines. In these experiments, each segment with composite delayed rewards was treated as an independent trajectory, and the modified algorithms were applied accordingly.

### A.2    IMPLEMENTATION DETAILS AND HYPER-PARAMETER CONFIGURATION

In our experiments, the policy optimization module was implemented based on soft actor-critic (SAC) (Haarnoja et al., 2018). We evaluated the performance of our proposed methods with the same configuration of hyper-parameters in all environments. The back-end SAC followed the JaxRL implementation (Kostrikov, 2021), which is available under the MIT License.

The proposed CoDeTr was built upon the GPT implementation in JAX (Frostig et al., 2018), available under the Apache License 2.0. Our experiments employed a Causal Transformer with three layers and four self-attention heads, followed by an in-sequence bidirectional attention layer with one self-attention head. For a comprehensive overview of the CoDeTr's hyper-parameter settings, please refer to Table 1.

Table 1: Hyper-parameters of CoDeTr.

| Hyper-parameter | Value |
|---|---|
| Number of Causal Transformer layers | 3 |
| Number of in-sequence attention layers | 1 |
| Number of attention heads | 4 |
| Embedding dimension | 256 |
| Batch size | 64 |
| Dropout rate | 0.1 |
| Learning rate | 0.00005 |
| Optimizer | AdamW (Loshchilov & Hutter, 2018) |
| Weight decay | 0.0001 |
| Warmup steps | 100 |
| Total gradient steps | 10000 |

For the baseline methods, the IRCR (Gangwani et al., 2020) method was implemented following the descriptions provided in the original paper. Both the RRD (Ren et al., 2021) and LIRPG (Zheng et al., 2018) methods are distributed under the MIT License. The code for HC (Han et al., 2022) is available in the supplementary material at `https://openreview.net/forum?id=nsjkNB2oKsQ`, while the code for RBT (Tang et al., 2024) can be found in the original paper.

To maintain consistency in the policy optimization process across all methods, each was subjected to 1,000,000 training iterations. For the proposed method, a dataset of 10,000 time steps was first gathered to pre-train the reward model. This model underwent 100 pre-training iterations, which

was deemed necessary to properly initialize the reward model before commencing the main policy learning phase. After this warm-up period, the reward model was updated for 10 iterations following the addition of each new trajectory. Furthermore, to monitor performance systematically, evaluations were conducted every 5,000 time steps. During prediction, the sequence length used for prediction is set to $H = 100$. All computations were performed on NVIDIA GeForce A100 GPUs with 40GB of memory, which were dedicated to both training and evaluation tasks.

### A.3 DATA NORMALIZATION PROCEDURES

The normalization process for our data varies depending on the type of composite delayed reward. Specifically:

- For **SumSquare**, the normalization is calculated as:

$$\frac{\sum \hat{R}_{\mathrm{co}}}{T \cdot \sum (r_{\max})^2}.$$

- For **SquareSum**, the normalization is given by:

$$\frac{\sum \hat{R}_{\mathrm{co}}}{\sum \left( \frac{T}{n} \cdot (r_{\max} \cdot n)^2 \right)}.$$

- For **Max**, the normalization is computed as:

$$\frac{\sum \hat{R}_{\mathrm{co}}}{\sum r_{\max}}.$$

Here, $\hat{R}_{\mathrm{co}}$ represents the predicted composite delayed reward, $T$ is the total number of time steps in a trajectory, $r_{\max}$ is the maximum possible reward in the environment, and $n$ is the number of delayed steps in each segment.

In essence, the normalization process involves scaling $\sum R_{\mathrm{co}}$ by the maximum achievable reward in the given environment. This approach ensures that results from experiments with different delayed steps are on the same scale, enabling meaningful comparisons across varying delayed steps and their impact on the learning process.

## B ALGORITHM

---

**Algorithm 1** Policy Optimization with CoDeTr

---

1: **Initialize:** replay buffer $\mathcal{D}$, CoDeTr parameters $\psi$, and policy $\pi$.
2: **while** training is not complete **do**
3:     Collect a trajectory $\mathcal{T}$ by interacting with the environment using the current policy $\pi$.
4:     Store trajectory $\mathcal{T}$ with composite delayed reward information based on sequences $\{(\tau, R_{\mathrm{co}}(\tau))\}$ in replay buffer $\mathcal{D}$..
5:     Sample batches from replay buffer $\mathcal{D}$.
6:     Compute the mean squared error loss $(R_{\mathrm{co}}(\tau) - \hat{R}_{\mathrm{co}}(\tau))^2$ for CoDeTr using the sampled sequences from the replay buffer.
7:     Update CoDeTr parameters $\psi$ based on the computed loss.
8:     Relabel instance-level rewards in replay buffer $\mathcal{D}$ using the updated CoDeTr.
9:     Optimize policy $\pi$ using the relabeled data with an off-the-shelf RL algorithm (e.g., SAC (Haarnoja et al., 2018)).
10: **end while**

---

The training process involves alternating between updating the reward model and optimizing the policy, which creates a continuous loop of mutual improvement. First, the agent collects trajectories by interacting with the environment according to the current policy. These trajectories are then used to train the CoDeTr, which learns to predict instance-level rewards and composite delayed rewards for sequences. The training is done by minimizing the mean squared error (MSE) loss between

the predicted composite reward $R_{co}(\tau)$ and the observed composite delayed reward $\hat{R}_{co}(\tau)$. This loss function allows CoDeTr to accurately capture the relationships and dependencies within each sequence, ensuring that both individual and sequence-level contributions are effectively represented. Using the updated CoDeTr model, the rewards for state-action pairs in replay buffer are relabeled, providing more accurate feedback for policy optimization. The updated rewards are used to further refine the policy using reinforcement learning algorithms like SAC, enabling the agent to learn effective strategies even in environments with delayed rewards. This iterative procedure enhances both the reward model and the policy through each training cycle.

## C  ADDITIONAL RESULT

Table 2: Performance comparison across different settings, utilizing various delayed steps ranging from 25 to 200 in the Ant-v2 environment, evaluated over 3 independent trials. The scores presented are normalized to ensure comparability across different configurations. The methods that demonstrated the best performance, along with those that were statistically comparable based on a paired t-test at a significance level of $5\%$, are highlighted in boldface for emphasis.

| Delayed Type | SAC | LIRPG | HC | IRCR | RRD | RBT | CoDeTr(ours) |
|---|---|---|---|---|---|---|---|
| Sum | 0.0004 (0.0002) | −0.1759 (0.0631) | 0.0025 (0.0058) | 0.03364 (0.0281) | 0.3327 (0.2095) | **0.5699** (**0.0162**) | **0.5493** (**0.0187**) |
| SumSquare | −0.0067 (0.0022) | −0.0159 (0.0027) | 0.0198 (0.0005) | −0.0617 (0.0273) | 0.0308 (0.0207) | 0.2821 (0.0703) | **0.3910** (**0.0431**) |
| SquareSum | −0.0902 (0.1031) | −0.0012 (0.0001) | −0.0280 (0.0275) | −0.1060 (0.0303) | 0.0575 (0.0173) | 0.0890 (0.0240) | **0.1992** (**0.0110**) |
| Max | 0.04093 (0.0065) | −0.1982 (0.0373) | 0.0108 (0.0103) | −0.0093 (0.0524) | 0.2193 (0.0416) | 0.4669 (0.1078) | **0.5318** (**0.0821**) |

In Table 2, the performance of different methods is compared across various composite delayed reward types: Sum, SumSquare, SquareSum, and Max, on the Ant-v2 environment. Our proposed method, CoDeTr, consistently performed well across all composite delayed reward configurations, either achieving the best results or performing comparably to the top baseline methods. In particular, CoDeTr showed strong performance under different composite delayed reward settings, demonstrating its ability to handle complex reward structures effectively. These results indicate that our approach is robust and adaptable, providing high-quality performance across a range of composite delayed reward scenarios.

## D  DISCUSSION

**Limitation.**  Our experimental results demonstrate that the proposed approach performs effectively across various types of composite delayed rewards, showing notable improvements over baseline methods. However, we also observed increasing difficulty in efficiently learning the policy as the delay length grew longer. This challenge arises because longer delays weaken the temporal connection between specific actions and their resulting outcomes, increasing uncertainty when attempting to determine which actions contributed to the observed reward. Consequently, the diminished ability to effectively assign credit to individual actions complicates the policy training process, leading to slower convergence and reduced overall performance in scenarios with extended delay lengths, as evidenced in our experiments.

**Future Direction.**  A key area for future work lies in addressing the challenges posed by longer reward delays. Our experiments have shown that increasing the delay length significantly complicates credit assignment to individual actions. To better capture long-range dependencies in such settings, future research could focus on developing advanced temporal credit assignment methods, such as improved attention mechanisms or memory-augmented neural networks. These techniques may enhance the model's ability to trace rewards back to responsible actions, even in situations with extended delays.

Expanding the use of composite delayed rewards to broader application scenarios represents another promising direction. Domains such as healthcare, autonomous vehicles, and industrial robotics often involve delayed and complex feedback that makes instance-level rewards impractical. Investigating how our proposed approach can generalize to these real-world applications would demonstrate its practical utility and robustness. Moreover, such exploration could help identify potential modifications required to adapt the framework to specific challenges, such as safety and real-time requirements inherent in these domains.

In addition, integrating human-in-the-loop feedback with composite delayed rewards could significantly enhance the learning process. Human evaluators often assign feedback based on pivotal events and use non-linear reasoning, which traditional reward models may fail to capture. Incorporating human feedback more directly, possibly through preference learning models aligned with composite delayed rewards, could improve the agent's ability to learn behaviors that align with human expectations. This approach would be particularly valuable in interactive environments where understanding human intent is crucial for the agent's success.

