# OpenReview forum: "Beyond Simple Sum of Delayed Rewards: Non-Markovian Reward Modeling for Reinforcement Learning"
_ICLR.cc/2025/Conference — ICLR 2025 Conference Withdrawn Submission_

### Official Review · Reviewer_oY1X · 2024-10-24

**Soundness:** 2
**Presentation:** 2
**Contribution:** 2
**Rating:** 5
**Confidence:** 4

**Summary:**

This paper generalizes approaches for RL with delayed rewards by relaxing two key constraints: Markovian reward and Simple additivity of stepwise rewards. The proposed architecture, Composite Delayed Reward Transformer (CoDeTr), provides a sequential reward model learning without imposing specific reward structures. Numerical results show that CoDeTr performs promisingly in various delayed reward environments.

**Strengths:**

- This paper is easy to read and follow.
- Relaxing Markovian reward and simple additivity assumptions gives flexibility to a reward model learning,  possibly widening applicability in multiple cases of RL with delayed rewards.
- I appreciate the authors writing the reproducibility statement to guarantee their work's reproducibility. Not all papers submitted in this venue explicitly describe their reproducibility.

**Weaknesses:**

1. **Discussion on the non-causality of weight calculation in Equation (7) is required.**
- 1-1. The proposed architecture in Figure 2 uses a causal transformer for query, key, and $\hat{r}$. However, in equation (4), I do not understand why $w$ incorporates state-action pairs up to $i+n_i-1$, not $t$ as in the case of the query, key, and $\hat{r}$, which means that the learned instance-level rewards include future information. However, is it valid to assume that the current reward includes future information?
- 1-2. I recommend using causal mixing in equations (6) and (7) (i.e., use $k_{t'}$ for $t' \leq t$) and report the result as a part of the experiment section. (In this case, the weight calculation in (7) can adopt averaging over $i \geq t$ for each $t$, not summing over all $i$).
- 1-3. Plus, there is no $t$ in equation (7). I guess the authors unintentionally drop a subscript of $t$ by mistake.

2. **Ablation study is required on core components: non-Markovian reward modeling and weighted sum.**

- 2-1. The authors should report results by ablating the weighted sum part (e.g., use uniform weight instead) for both the sum-form and general composite form.
- 2-2. The authors should report results by ablating non-Markovian reward modeling for both sum-form and general composite form that can be done by (i) ablate causal modeling of query and key, (ii) ablate causal modeling of $\hat{r}$, and (iii) ablate both. Here "ablate" means not using the causal transformer and using MLP stepwise.
- 2-3. Is there any specific reason why equation (5) uses linear transformation, not a non-linear one? Can non-linear compression improve performance?

3. **The novelty is not significant compared with a baseline (namely RBT)**
- One of the proposed algorithm's two key components, modeling non-Markovian reward, appears to mirror the RBT algorithm closely. A main difference is the adaptive weight learning using softmax operation, but RBT also adopts softmax operation for the instance-level reward calculation. The authors should clarify the specific innovations over RBT more explicitly to guarantee their novelty.


4. **Readability should be improved.**

- 4-1. The authors should improve the motivation for the delayed reward setting. Is there any concrete example other than the human annotation perspective?
- 4-2. Section 2.3 should be included in Section 3 since CoDeMDP is the authors' proposed concept.
- 4-3. Placing the pseudocode in the main body can help readers' understanding for the algorithm.

5. I recommend increasing the number of seeds to more than 3 (at least 5).

**Questions:**

Please see the weaknesses part above.

---

### Official Review · Reviewer_w2rF · 2024-10-28

**Soundness:** 2
**Presentation:** 3
**Contribution:** 2
**Rating:** 3
**Confidence:** 4

**Summary:**

This paper focuses on learning reward functions in settings where rewards are delayed.
The authors note that standard methods in this area assume (1) the existence of underlying Makov rewards and (2) delayed rewards are a sum of individual time step rewards. To that end, this paper proposes the Composite Delayed Reward Transformer, which allows for reward models that are non-Markovian and are the weighted sum of individual-level rewards.

**Strengths:**

1. The paper is well written and easy to follow.

2. Finding which states/actions are significant and should be assigned higher weight is an important problem not only in the delayed reward setting but also in the preference-based RL setting.

3. The addition of the Case Study was interesting and provided further insight into the quality of the proposed algorithm.

4. The authors compared their algorithm against an extensive set of related baselines.

**Weaknesses:**

Similarity with previous work:

This paper is extremely similar to Reinforcement Learning from Bagged Reward [1]. Both algorithms use transformer networks for the instance-level (per time step) reward prediction, where the attention mechanism operates on the sequences corresponding to the rewards. The algorithm's (and the pseudocode's) training process is also very similar.
In addition, both works assume non-Markovian rewards, even though the authors explicitly mention in the text that methods in this domain assume Markovian rewards.

The authors should clarify how their work differs from [1].
[1] https://arxiv.org/pdf/2402.03771


Missing related work:

The authors should mention the sub-field of preference-based RL, which can learn non-markovian reward models from sequences of (state, action) pairs, especially because the authors mention human feedback and evaluation in the introduction.


Unclear on Experimental Design Choices:

1. The authors decided to use 6 different reward delay lengths but elected not to run experiments in the truly sparse setting, where the agent receives a reward at the end of the episode. This seems surprising because the sparse reward setting is a commonly studied problem.

2. Why did you include these specific composite delayed reward structures, SumSquare, SquareSum, Max?

3. SumSquare is supposed to be the sum of the squared step rewards, but why is the formula the sum (abs(r)*r)? Shouldn’t it be (r**2)? If the reward is negative, then abs(r)*r is not the same as (r**2)? The same issue occurs in the SquareSum. Can the authors explain this?

Overclaiming on experimental results:

The authors claim their algorithm consistently outperforms baseline methods across different environments (L427). However, reviewing Figure 3, CoDeTr seems comparable to the other baselines. It would be helpful if the authors clearly specified in how many experiments their algorithm outperformed all baselines. In addition, running a statistical analysis (i.e., t-test) on these results would provide further support for any claims.

**Questions:**

1. It is unclear how contributions 1 and 2 are different.

2. Definition 1 is the definition of a finite MDP. This should be included in the description.

3. In Definition 1, why was no discount factor included?

4. In Figure 5, are the observed and predicted rewards supposed to be similar?

5. In Figure 5, why doesn’t the observed delayed reward drop when the agent's performance fails?

**Details Of Ethics Concerns:**

The paper is really similar to [1], not only in the algorithm but also the writing itself.

In particular, the problem formulation in [1], Section 3.2 is very similar to the problem formulation presented in this work, Section 2.3 (Reinforcement Learning from Composite Delayed Reward).

In addition, Section 4.2, paragraph Causal Transformer for Reward Representation in [1] reads almost identical to Section 3.2, paragraph Instance-Level Reward Prediction from this submitted paper.

[1] https://arxiv.org/pdf/2402.03771

---

### Official Review · Reviewer_M6XV · 2024-10-30

**Soundness:** 3
**Presentation:** 3
**Contribution:** 2
**Rating:** 3
**Confidence:** 3

**Summary:**

The paper studies a delayed reward setup, where an RL agent observes only delayed rewards summarizing transition sequences rather than the true Markovian rewards. The main challenge addressed is when these delayed rewards are defined as non-linear transformations of the sequence of true rewards. The authors introduce the CoDeMDP framework to formalize the problem and address it by using a Transformer model, CoDeTr, which decomposes delayed rewards into state-action rewards that can then be used to train any standard RL algorithm. The experiments demonstrate that the proposed solution improves upon existing baselines when the delayed reward is defined as a non-linear function of the true rewards and performs comparably to the baselines when the delayed reward is simply a sum.

**Strengths:**

The paper is well-structured, presenting the challenges it addresses and the proposed solution clearly. The experiments section is well-organized and Apenndices are comprehensive.

**Weaknesses:**

While the paper is well-structured and tackles an interesting challenge, I have some reservations about the importance of its contributions, mainly due to these two points:

1. The problem of non-Markovian or weighted delayed rewards feels somewhat artificial. Beyond the rather vague examples in the introduction, it’s unclear in what specific tasks or RL environments this problem would naturally occur. Without clearer context, the paper seems to be addressing a problem it has introduced.

2. CoDeTr decomposes delayed rewards into per-transition rewards, but these per-transition rewards are still non-Markovian, and this limitation seems unaddressed.

Due to the first point, the practical relevance of the proposed solutions is unclear, and the second point raises doubts about its theoretical value.

That said, I would consider recommending acceptance if these concerns are addressed in the authors' response.

**Questions:**

1. Why do you sum over all subsequences $\tau\subset \mathcal{T}$ in Eq. (3) ?

2. As I understand, the RL agent is trained in an otherwise standard MDP, with rewards defined as $r_t=w_t(\psi)\hat{r}_t(\psi).$ Is that correct? If so, I believe this should be stated explicitly. Besides, this reward is nan-Markovian; please, comment on that.

3. What would be the effect of using a non-fixed number of delayed steps? This seems more realistic for any potential applications that come to mind.

4. When and how could the proposed method/ideas benefit the researchers and/or practitioners?

---

### Official Review · Reviewer_kvYG · 2024-11-08

**Soundness:** 2
**Presentation:** 4
**Contribution:** 3
**Rating:** 5
**Confidence:** 4

**Summary:**

This paper proposes the CoDeTr architecture to address the credit assignment problem in a delayed reward environment. CoDeTr is a Transformer that learns a scalar compound reward and extracts the reward at each step through a in-sequence attention mechanism.

**Strengths:**

[S1] The model focuses on scalar reward prediction, simplifying the loss function compared to RBT, which uses additional reward bags and state prediction errors, yet demonstrates superior performance.

[S2] CoDeTr’s design, with whole-sequence Transformer-based reward summation, accommodates non-Markovian rewards and variable step importance, making it potentially more effective in such environments.

**Weaknesses:**

[W1 & Q1] Reward and weight validity

Without additional constraints, it is unclear if $ r $ and $ w $ outputs genuinely reflect rewards and weights.
Since $ R_{co} $ is a weighted sum of $ r $ and $ w $ outputs, additional experiments could help clarify their interpretive accuracy.

[W2 & Q2] Performance in environments without actual per-step reward

The experiments are conducted on environments with known per-step rewards, artificially accumulated as composite rewards.
Testing on complex environments with no per-step reward signals, such as robotics tasks with task-completion as the final reward or RLHF for language models with only a final reward, would enhance the assessment of CoDeTr’s performance and applicability in broader contexts.

[W3 & Q3] Impacts of model expressiveness

In Figure 4, the best two methods appear to be CoDeTr (proposed) and RBT. They both use a full-sequence Transformer model for credit assignment, while the other methods use a two-layer MLP. The performance of CoDeTr and RBT is also very close.
The full-sequence Transformer model takes in the entire sequence as input and is more powerful. However, the MLP only takes in the current Markovian state/action as input and is less expressive.

**Questions:**

[Q1-Q3] See Weaknesses

[Q4] Open question: If r and w outputs are not actually predicting rewards and weights, what's the effect on downstream RL learning?

---

### Note · Authors · 2024-11-24

**Comment:**

Thank you for your valuable suggestions. We will revise our paper further based on your feedback.

Regarding the ethical concerns raised by Reviewer w2rF, we would like to clarify the distinctions between our work and [1]. Our approach generalizes the framework of [1] to address a broader class of reward structures. Although both works use two-layer Transformer architectures, their roles are fundamentally different: in [1], the attention layer is integrated into the reward model, while in our work, it is used outside the reward model to learn instance-specific weights. Any similarities in phrasing are coincidental and stem from the technical nature and shared terminology of the field, not from duplication. We hope this clarification addresses the ethical concerns.

[1] https://arxiv.org/pdf/2402.03771

**Withdrawal Confirmation:**

I have read and agree with the venue's withdrawal policy on behalf of myself and my co-authors.